# Three-Dimensional Analysis of the Pulp Chamber and Coronal Tooth of Primary Molars: An In Vitro Study

**DOI:** 10.3390/ijerph19159279

**Published:** 2022-07-29

**Authors:** Montserrat Diéguez-Pérez, Jesús Miguel Ticona-Flores

**Affiliations:** 1Preclinical Dentistry Department, Faculty of Biomedicine and Health Sciences, European University of Madrid, Villaviciosa de Odón, 28670 Madrid, Spain; 2Faculty of Biomedicine and Health Sciences, European University of Madrid, Villaviciosa de Odón, 28670 Madrid, Spain; jesus.ticona.f@upch.pe

**Keywords:** primary teeth, dental pulp cavity, dental pulp anatomy, cone beam computed tomography

## Abstract

The anatomical variability of primary molars promotes research to improve clinical restorative activity, forensic identification, and industrial development. The objective was to study the morphology of the pulp chamber and its three-dimensional relationship with the external morphology of the tooth. A total of 151 deciduous molars were collected and subjected to three-dimensional imaging analysis in order to determine dental crown (CV) and pulp chamber (PV) volumes, their ratio (VR), pulp chamber roof (PCR) and pulp chamber floor (PCF) area, the furcation length (FD), and morphological peculiarities. The data were compared using the Kruskal–Wallis test in SPSS 26 software. The statistical analysis determined statistically significant differences between the four groups of molars for all variables. Great anatomical variability was observed, especially in the maxillary first molar, the volumes were higher in the maxillary second molar and the highest risk of furcal perforation was seen in the mandibular first molar. Additionally, mandibular second molars with six pulp horns, and some different density images compatible with accessory canals and pulp stones were found. Based on the findings of this investigation, we confirm the great anatomical variability that exists between the maxillary and mandibular molars.

## 1. Introduction

The preservation of the pulp chamber of primary teeth is clinically determined by the knowledge of its variations, proportions, and spatial relationship with the rest of the hard tissues of dental crown [1,2,3]. This fact becomes more important in a primary tooth, as it presents a lower thickness of enamel and dentine tissues [4]. Making a correct access cavity in pulp therapy allows for the removal of all tissues housed in the chamber and correct visualisation of both the pulp floor and the entrance to the root canals, thus improving the long-term prognosis [5,6,7,8,9]. Furthermore, knowledge of the morphological characteristics of the chamber pulp floor prevents possible iatrogenesis during the technique.

Morphological aspects, such as accessory canals or the permeability of the floor of the pulp chamber, are also important to consider in the failure of pulpal therapy because the presence of these facilitates the bacterial contamination of pulpal necrosis to the interradicular bone and causes pathological resorption [10,11,12,13].

The three-dimensional analysis allows more efficient detection and localization of mineralised tissues within the connective tissue of the pulp chamber, which generally has a nodular appearance [14]. Furthermore, the primary teeth anatomy study has a legal and forensic application since knowledge of the length of the pulp and coronal cavity in mandibular primary molars could be a simple and non-invasive analysis that allows for an estimate of children’s dental and chronological age [15].

At present, the researchers focus on advanced imaging techniques, such as cone beam computed tomography (CBCT) [16,17,18,19,20,21,22,23], micro computed tomography (micro CT) [1,2,3,4,5,19,24,25,26,27], and spiral computed tomography (SCT) [9]. A three-dimensional reconstruction is possible, reflecting greater accuracy of the most relevant anatomical features of the pulp chamber, thus being able to explain the successes and failures of different restorative or pulp therapies. However, there is a lack of research on this topic because most studies focused on one racial group [1,2,5,6,13]. For this reason, as of today, despite the studies, knowledge of the deciduous teeth anatomy is still insufficient due to the great anatomical variability it presents.

Five pillars motivate our interest and justify research in this field: the great anatomical variability in primary teeth; the little research on these anatomical peculiarities; the need for greater morphological knowledge of the tooth for the clinical success of pulpotomy treatments, the demand of which increased to maintain the integrity of the primary teeth until the replacement of permanent teeth; the increase in the restoration of primary teeth with aesthetic preformed crowns, a technique that allows small changes, thus making it difficult to adapt to the mouth; and its applicability in forensic dentistry. Therefore, the main objective was to study the morphology of the pulp chamber and its three-dimensional relationship with the external morphology of the tooth.

## 2. Materials and Methods

This in vitro and observational study was approved by the Ethics Committee of the San Carlos Clinical Hospital in Madrid (Spain) with internal code 21/375-E.

### 2.1. Sample Selection

The first and second deciduous molars preserved their occlusal anatomy, where pits, fissures, and grooves could be distinguished. Grade 0 (non-visible wear) on the “Tooth Wear Evaluation System” [28] was collected from different dental offices regardless of age or sex. Teeth extracted due to orthodontic indication or exceeding their normal exfoliation time were included. Additionally, those teeth that presented with fillings, trauma, pulp treatments, anomalies of the dental structure, caries, or exposed dentine were excluded.

It must be highlighted that: only the teeth of children whose parents signed the donation consent were collected.

For the sample calculation, a bilateral hypothesis, power of 80%, and the significance of *p* < 0.05 were established. Furthermore, Orhan’s research was taken as a reference [3] that compared chamber pulp volumes in three-dimensional reconstructions of boys and girls (77 SD ± 4 mm^3^ and 64 ± SD 5 mm^3^), resulting in a sample size of at least 40 molars.

### 2.2. Storage and Group Formation

The collection of teeth was protocolised, so that immediately after the exodontia, the blood remains were removed and placed in a plastic bottle with physiological saline solution until transported to the analysis centre.

Subsequently, the teeth were sanitised with 4% chlorhexidine to remove organic tissues, disinfected with 3% sodium hypochlorite [29], and finally stored in a 9% sodium chloride solution at a temperature of 4 °C. until the moment of taking 3D images.

Four working groups were established: the deciduous maxillary first molars (P1MS); the deciduous mandibular first molars (P1MI); the deciduous maxillary second molars (P2MS); and the deciduous mandibular second molars (P2MI).

### 2.3. Image Acquisition and 3D Analysis

To fix the teeth, they were mounted on heavy silicone blocks for subsequent CBCT scanning. The images were obtained with CS 8100^®^ tomographic equipment (Carestream Dental, Atlanta, GA, USA), with a voltage of 90 kV, a tube current of 15 mA, and a slice thickness of 0.75 μm.

For the analysis of the images, the CS 3D Imaging Light version 3.10.22.0 (Carestream, Rochesta, NY, USA) and 3D Slicer version 4.11.202110226 software [30] were used.

In the 3D Slicer software, the “Threshold” tool was used to calculate the Houndsfield units of each dental structure, which were used for the three-dimensional reconstruction of the crown, cervical third of the root, and the surfaces of the pulp chamber roof and floor (Figure 1). Finally, the variables were measured using the “Quantification” tool that calculates the volume or area depending on the reconstructed structure.

### 2.4. Measurements

#### 2.4.1. Pulp Chamber Roof (PCR) and Pulp Chamber Floor (PCF)

The pulp chamber roof (PCR) was reconstructed, in blue, using the Houndsfield units of the first layer of dentine parallel to the occlusal surface that covered the pulp chamber and delimited by the pulp horns.

The surface of the pulp chamber floor (PCF) was reconstructed, in green, using the Houndsfield units of the first layer of dentine that form the bifurcation area of the roots, close to the neck of the tooth, without entering the root canals.

Both surfaces were measured in mm^2^ by the “Quantification” tool of the 3D Slicer software (Figure 2).

#### 2.4.2. Distance between Pulp Chamber Floor and Furcation (FD)

To measure the distance between the pulp chamber floor and the root bifurcation (FD), a line was drawn from the most convex spot on the pulp chamber floor to the most concave spot on the root bifurcation. This line was measured with the “line” tool in the 3D Slicer software in mm (Figure 3).

#### 2.4.3. Pulp Chamber Volume (PV), Dental Crown Volume (CV), and Volume Ratio (VR)

The anatomical crown was selected, which was delimited at the cervical level by the limit of the cement-enamel junction. This selection was called a dental crown. Using the “Quantification” tool, the dental crown volume (CV) and pulp chamber volume (PV) were calculated in mm^3^.

The volume ratio (VR) was calculated by dividing the volumes of the pulp chamber and the dental crown. This measure was expressed as a percentage (Figure 4).

#### 2.4.4. Other Peculiarities

After the described measurements, the internal morphology of the coronal segment was analysed to study the presence or absence of atypical and unusual anatomical details using CS 3D Imaging Light software version 3.10.22.0 (Carestream, Rochesta, NY, USA).

The presence of hyperdense images within the pulp chamber compatible with a pulp stone, whose presence could be corroborated in three-dimensional reconstructions, was analysed.

Additionally, hypodense images within the pulp chamber floor, compatible with accessory canals, were also observed. These should be seen in the axial view of the CBCT as a line or point with less density than the structure of the hard tissue. Their presence was corroborated by observing the same hypodensity image in the coronal and sagittal views [31] (Figure 5).

### 2.5. Calibration, Pilot Test, Internal Validity

The measurements were made by a specialist in paediatric dentistry, who was trained by a specialist professor in maxillofacial radiology, for the CBCT observation, interpretation, and three-dimensional reconstruction software. The inter-observer concordance was evaluated using the kappa coefficient, 0.98 (95%CI 0.97–0.99).

The researcher in charge of the data collection did a pilot test with 26 teeth to verify the feasibility, standardization, and reproducibility of the data collection process, 15 days after the whole measurement process was carried out again, from the three-dimensional reconstruction of the root to taking measurements such as coronal and pulp chamber volume; pulp chamber roof and floor surface; and the pulp chamber floor to the root bifurcation length; in order to assess the intra-observer concordance through the interclass correlation coefficient (ICC). Obtaining as ICC results for CV 0.98 (95%CI 0.97–0.99); PV 0.98 (95%CI 0.97–0.99); PCR 0.98 (95%CI 0.95–0.99); PCF 0.98 (95%CI 0.96–0.99); and FD 0.92 (95%CI 0.85–0.97).

### 2.6. Statistical Analysis

The IBM SPSS vs. 26 software was used for the statistical analysis. Descriptive statistics, such as frequency, mean, and standard deviation were used. The significance of *p*-value < 0.05 and a power of 80% was set. The normal distribution was confirmed with the Shapiro–Wilk test (*p* < 0.05). Therefore, the non-parametric Kruskal–Wallis test was used for the inferential statistics for quantitative variables and the chi-square test to compare qualitative variables.

## 3. Results

### Sample Distribution

The number of teeth collected was 151, after excluding 19 due to dental caries, 2 with dental trauma and 26 due to excessive occlusal wear. Finally, 104 teeth were analysed, in their distribution: 27 primary maxillary first molars (P1MS), 29 primary mandibular first molars (P1MI), 22 primary maxillary second molars (P2MS), and 26 primary mandibular second molars (P2MI).

The averages of each length and dimension per tooth, as well as the ratio established between the soft and mineralised tissue at the coronal level, are reflected in Table 1 and Figure 6.

A statistically significant difference was observed between the four groups and related study variables using the Kruskal–Wallis test for each measurement performed (Table 1).

The limits obtained at the 95% confidence interval concerning the total surface of the pulp chamber roof were 24.44 and 27.69, respectively, in the P1MS, 26.46, and 30.46 in the P1MI, 3.51 and 41.10 in the P2MS, 39.17 and 44.59 in the P2MI (Figure 7a). The limits obtained at the 95% confidence interval for the total surface of the floor of the pulp chamber were 9.85 and 11.12, respectively, in the P1MS, 7.72 and 9.23 in the P1MI, 15.74 and 18.94 in the P2MS, 10.47 and 12.07 in the P2MI (Figure 7b).

The limits obtained at the 95% confidence interval, regarding the length of the pulp chamber floor to the furcation, were 0.92 and 1.13, respectively, in the P1MS, 0.74 and 0.88 in the P1MI, 1.10 and 1.35 in the P2MS, 1.16 and 1.37 in the P2MI (Figure 8).

The limits obtained at the 95% confidence interval regarding the volume of the pulp were 20.76 and 25.69, respectively, in the 1MS, 24.33 and 29.57 in the P1MI, 38.82 and 50.03 in the P2MS, 35.55 and 48.82 in the P2MI (Figure 8). Regarding the coronal volume, the established limits were 165.07 and 182.30, respectively, in the P1MS, 175.85 and 191.66 in the P1MI, 343.68 and 376.12 in the P2MS, 323.36 and 353.38 in the P2MI (Figure 9).

The limits obtained at the 95% confidence interval concerning the proportion between pulpal and coronal volumes were 24.44 and 27.69, respectively, in the P1MS, 26.46 and 30.46 in the P1MI, 37.51 and 41.10 in the P2MS, 39.17 and 44.59 in the P2MI (Figure 10).

In the molars studied, six accessory pulp horns were visualised, five (19.23%) located distal to the distolingual horn of the P2MI, and one (3.70%) located distally to the palatine horn of the maxillary first molars (Figure 11).

The hyperdense images, compatible with the pulp stone observed and that could be reconstructed in three dimensions, had a frequency in the P1MS of five (18.52%), of which, two (7.41%) were located in the coronal third, and three (11.11%) in the middle third. In the P1MI, two (6.90%) images were visualised, both located in the middle coronal third. In the P2MS, three (13.64%) were observed, two of them in the coronal third (9.09%) and one in the cervical third (4.55%). Finally, in the P2MI, two images were detected in the middle third (7.69%) (Figure 12).

When the tomographic views were analysed, hypodense images (accessory canals) were observed on the pulp chamber floor, which were compatible with accessory canals. Its frequency by molar was 12 (44.44%) in the P1MS; 9 (40.91%) in the P2MS; 10 (34.48%) in the P1MI; and 12 (46.15%) in the P2MI. To statistical contrast with the chi-square test, no statistical significance was found (*p* > 0.05) between the presence and absence of these peculiarities in the different dental groups. Furthermore, hypodense images on the pulp chamber roof were analysed. Its frequency was 3 (11.11%) in the P1MS; 6 (27.27%) in the P2MS; 4 (13.79%) in the P1MI and 11 (42.31%) in the P2MI.

## 4. Discussion

### 4.1. Study Characteristics and Morphometric Analysis

Traditionally, dental dimensions are obtained from study models, extracted teeth measured with a calliper or through an X-ray and a millimetric template [7,13,32,33]. However, it is not common to find studies that reflect these measurements three-dimensionally and accurately reflect the morphological characteristics of the primary tooth. For this reason, this research provides relevant data with a clinical and didactic application, such as the improvement regarding access to and removal of the pulp chamber roof, as well as adequate cavity preparation without the risk of damaging the pulp chamber floor. Additionally, the data provided could have applicability in the industry and for the manufacture of preformed aesthetic crowns.

Some studies have an “in vivo” methodology [7,8,15,19]. This provides specificity and reliability to the research, which is an advantage of this type of methodology. Nevertheless, for the analysis of biological effects, this methodological feature is irrelevant to determining the anatomical dimensions of a primary tooth; moreover, radiation in paediatric patients should be restricted to extremely necessary cases. However, the “in vitro” methodology is more rigorous in anatomical studies, such as this and other studies [1,2,3,4,5,6,9,11,12,13,16,27,34,35,36], by allowing the fixation of the teeth during handling.

Regarding the size of the sample, only one investigation surpasses ours in this aspect, which made measurements in bitewing radiographs, analysing 153 teeth [7], therefore, in two dimensions (2D). Other research studied a range of numbers between 5 and 100 molars [1,2,3,4,5,6,11,34,35,36].

### 4.2. Pulp Chamber Dimensions

No scientific evidence was found for 3D reconstruction studies that measure the surface of the pulp chamber roof and floor (mm^2^). The largest and smallest surfaces of the pulp chamber roof were observed in the P2MI (41.88 ± 6.70) and P1MS (26.07 ± 4.12), respectively, this is because these teeth have the largest and smallest number of both cusps and pulp horns. Regarding the size of the pulp floor, the P2MS (17.34 ± 3.61) is the one with the greatest extension and the P1MI (8.48 ± 1.99) the one with the smallest surface, this results from the fact that the maxillary molar is the largest tooth, with three roots, while the mandibular first molar is the smallest tooth with two roots.

### 4.3. Furcation Dimensions

Furcation length analysis showed that the tooth with the lowest risk of perforation was P2MI (1.26 ± 0.26), followed by P2MS (1.22 ± 0.27), P1MS (1.02 ± 0.26), and finally the P1MI (0.81 ± 0.17). It is highlighted that the mandibular teeth represent, simultaneously, a lower and a higher risk of pulp failure; the reason for this peculiarity is unknown, it could be due to such a difference between the teeth of both maxillaries. Dabawala et al. [7] opted for two-dimensional analysis of bitewing radiographs and radiographic grids (mm), for this reason, no comparison between our data and this study was possible. However, a qualitative evaluation showed that the greatest distance between the floor of the pulp chamber and the bifurcation was the P2MS (1.78 ± 0.25), followed by the P2MI (1.76 ± 0.39) and the P1MS (1.70 ± 0.38), a different order from our data. The highest risk of perforation at this level is presented by the P1MI (1.59 ± 0.31), which coincides with our results. On the other hand, for Selvakumer et al. [9], the distance between the most convex spot of the pulp chamber’s floor to the root bifurcation was greater in P1MI (1.98 ± 1.88 mm) than in P2MI (1.88 ± 0.60 mm), results that differ between our results and those of other researchers [7]. 

### 4.4. Dental Crown and Pulp Chamber Volume

The largest coronal and pulpal volumes were observed in the P2MS, and although this tooth has fewer cusps and pulp horns compared to the P2MI, the larger volume of the P2MS may be due to the presence of Carabelli’s tubercle and the greater dimension in the buccolingual direction [4]. There is no doubt that the smaller volumes of both structures (PV and CP) were found in the first molars [4], although to study the size, some authors reported mesiodistal and buccolingual linear measurements in mm [7,32]. Reporting that the greatest measure was found in the P2MS and buccolingual direction [32], in addition to a greater buccolingual length of P1MI than the P1MS [4,32]; contradictory data to our study, where the smallest dimensions (mm^3^) corresponded to P1MS. Orhan et al. [3] determined the mean pulp volume of maxillary molars, taking gender into account. It was 77 mm³ ± 4 in boys and 64 mm³ ± 5 in girls. These data, although not comparable, are far from our results.

The highest ratio between the pulp and crown in the first molars was observed in P1MS (7.93 ± 1.97), while in the second molars, P2MI presented the highest ratio. Amano et al. [1] studied this proportion in 10 maxillary second molars, finding a mean of 9.6 ± 0.5 in five young teeth and 8.1 ± 0.6 in five teeth that remained in the mouth longer; our result is reflected in the range obtained by these authors (8.68 ± 2.25). Ikari et al. [2] also analysed this proportion in 20 mandibular second molars at different stages of the dentition and obtained a range of 6.8–8.8 during the mixed and deciduous dentition period, respectively. Our results (8.77 ± 2.32) for these molars are superior to these researchers. Orhan et al. [3] determined this proportion in maxillary molars and differentiated only between boys and girls, their results reflect a higher mean in males (7.20) compared to girls (6.80), data not comparable with those of our study, but could be a relevant factor for consideration in future research. Knowledge of the dimensions of these variables allows the clinician to access the root canals with a lower probability of furcal perforation.

### 4.5. Anatomical Peculiarities

Regarding the anatomical peculiarities, Agematsu et al. [5] observed five pulp horns in the mandibular first molar. However, in this study, six pulp horns were found, five of which are located distal to the distolingual pulp horn of the mandibular second molars, these being the smallest. The frequency of the sixth pulp horn in this position reached 38.5% of the sample size. This factor can undoubtedly affect the extent of the cavity access for coronal pulp removal. Furthermore, a hyperdense image only was observed at the entrance of the root canal compatible with canal calcification or a pulp stone; some researchers [37] located them in the tooth cervical third, presenting bilaterally in the same patient.

Twelve hyperdense images were observed in the pulp chamber, compatible with pulp stones, the majority were found in the P1MS. The most frequent location of the possible pulp stone was the middle of the coronal third, a fact to be considered when planning pulpal therapy. However, scanning electron microscopy (SEM) is the correct technique to determine the presence and morphology of these pulp stones. It is the high cell growth rate of the coronal pulp that is the cause of this nodular mineralisation [14].

The study of accessory canals at the furcation is more appropriate with the scanning electron microscopy technique, Kramer et al. [11] observed that the external furcation area showed a higher prevalence of accessory canals concerning the internal zone. Likewise, Kumar et al. [38] noted a greater number of accessory canals in the maxillary molars, these canals were larger and varied morphology (spherical and oval), the spherical canals being the most prevalent.

The high prevalence of accessory canals were described. Authors such as Lugliè et al. [12] found these canals in the 77% of the deciduous molars furcation, data similar to that reported by Sharma et al. [13] (73.3%), and Wrbas et al. [36], who found a prevalence of accessory canals in the maxillary second molars of 80% and 75% in the mandibular second molars, of which a low percentage (17.3%) were located in the floor of the pulp chamber and 82.7% in the external furcation area, near the periodontal ligament. 

Dammaschke et al. [35] observed that 94% of the accessory canals in primary teeth had diameters of 200 μm, approximately. This fact reflects the great risk of failure in the pulp therapy of primary teeth because accessory foramina with large diameters favour the spread of an inflammatory process from the pulp to the periodontal tissues.

In this study, the CS 3D Imaging Light software was used to analyse the presence of these canals. The fissures observed at this level could be the result of tooth desiccation during handling in the tomographic technique. Its presence in almost half of the sample (41.35%), would indicate numerous visible accessory canals by conventional tomography. Alternatively, the presence of these fissures on the surface of the pulpal roof and in a coronal direction (23.08%) makes us suspect the great permeability that exists at this level and the ease with which microorganisms have access to the chamber pulp, thus favouring the risk of inflammatory and infectious processes.

### 4.6. Limitations and Strenghts

A limitation of this study was the lack of records regarding the tine of tooth eruption, as the amount of mineralised tissue and the pulp volume can vary according to the degree of apposition of secondary dentine that occurs when the tooth ages, significantly influencing the tooth/pulp ratio. [1,2] Additionally, it should be considered that primary teeth in mixed dentition have less enamel thickness due to attrition [1], a factor that was considered during the selection of the sample. Neither was the sex of the patient considered, a factor that could affect the coronal and pulpal dimensions [32,33], although there is no evidence of this in three-dimensional analysis.

Despite these limitations, such as the thickness of the hard and soft tissues at the coronal level in primary teeth, with the limited current literature on the matter, our results are able to facilitate professional clinical activity.

Most of the 3D investigations that reported on the ethnic characteristics of the study population corresponded to the teeth of Asian children [1,2,5,6,13], and this factor may affect the dental crown dimensions [6]. In this sense, we believe that our study provides valuable information, as it was conducted in a Caucasian population.

After analysing and comparing our results, we believe that the differences with other studies are: Firstly, most deciduous dentition anatomy studies are of Asian patients, unlike ours, who are of European children, and may have different tooth dimensions. Secondly, the most common methodology is the application of calliper or periapical X-ray, while CBCT and micro-CT are more contemporary. Finally, there are no studies considering the patient’s dentition stage at the time the primary molars were extracted

In forensic dentistry, coronal and pulp chamber heights were studied [12], using radiovisiography to determine whether the coronal index of the tooth was associated with the chronological age of the child. Based on this, a line of research can be opened to relate the pulp volume to the biological and chronological age of the child. In this research, it was decided not to collect the age of the patient; for this reason, age ranges were not established.

## 5. Conclusions

The three-dimensional analysis of the dental crown of posterior primary teeth demonstrates the great anatomical variability that exists between the first molars, especially with the volume ratio of the pulp chamber and dental crown.

The larger and smaller surfaces of the pulp chamber roof do not correspond to the size of the molar dental crown. Therefore, those with smaller crown dimensions are at a higher risk of pulpal pathology.

Traditionally, the mandibular second molar is the tooth with the largest coronal dimension, as it has the largest number of cusps and pulp horns; however, according to our study, the maxillary second molar is the one with the largest coronal and pulp volume.

The mandibular first molar is the tooth that presents the greatest risk of perforation at the furcation level when the pulpotomy technique is performed.

Some maxillary second molars have six pulp horns, an aspect to be considered during the removal of the chamber roof.

Most of the hyperdense images compatible with pulp stones are visualised in the maxillary first molars, which could affect the prognosis of some pulp therapies.

Hypodense images observed along the pulp chamber roof and floor favour the advance of cariogenic bacteria to the adjacent tissue.

## Figures and Tables

**Figure 1 ijerph-19-09279-f001:**
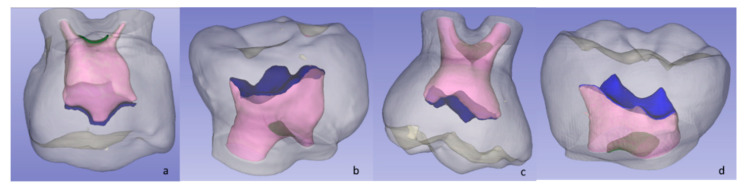
Three-dimensional reconstruction: (**a**) primary maxillary first molar (P1MS); (**b**) primary mandibular first molar (P1MI); (**c**) primary maxillary second molar (P2MS); and (**d**) primary mandibular second molar (P2MI).

**Figure 2 ijerph-19-09279-f002:**
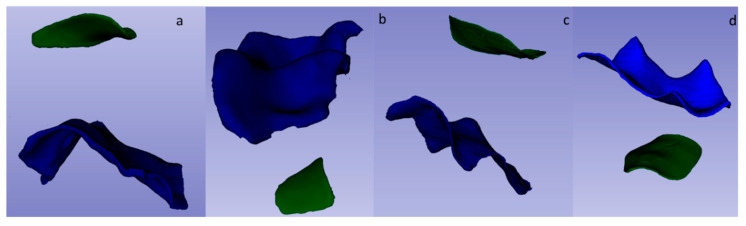
Three-dimensional reconstruction of the roof (in blue) and floor (in green) surfaces of the primary molar pulp chamber: P1MSP(**a**); P1MI (**b**); P2MS (**c**) and P2MI (**d**).

**Figure 3 ijerph-19-09279-f003:**
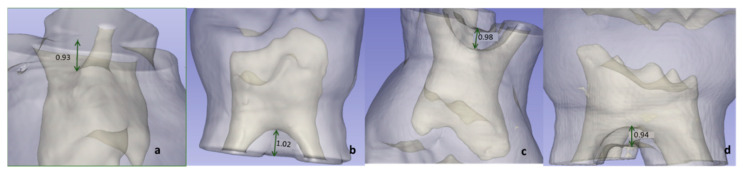
The green arrows indicate the measurement between the most convex spot from the pulp chamber floor to the root bifurcation (FD): P1MS (**a**); P1MI (**b**); P2MS (**c**) and P2MI (**d**).

**Figure 4 ijerph-19-09279-f004:**
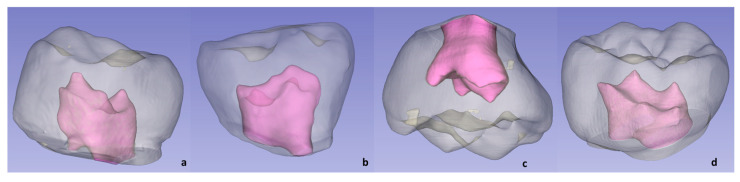
Three-dimensional reconstruction of primary molars’ teeth and pulp volumes: P1MS (**a**); P1MI (**b**); P2MS (**c**) and P2MI (**d**).

**Figure 5 ijerph-19-09279-f005:**
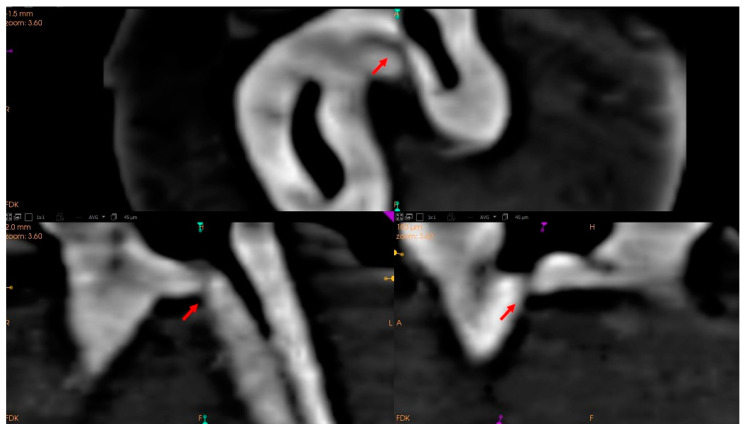
Tomographic views of the pulp chamber floor. Arrows point to hypodense images compatible with accessory canals. Axial view (**a**); coronal view (**b**); and sagittal view (**c**).

**Figure 6 ijerph-19-09279-f006:**
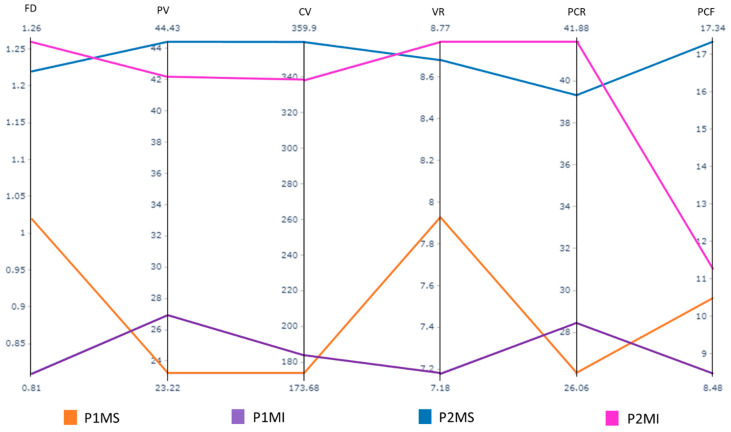
Parallel coordinates plot of study’s variables in which the means of each variable can be compared graphically. Pulp chamber roof (PCR); pulp chamber floor (PCF); distance between pulp chamber floor and furcation (FD); pulp chamber volume (PV); dental crown volume (CV); volume ratio (VR); primary maxillary first molar (P1MS); primary mandibular first molar (P1MI); primary maxillary second molar (P2MS); and primary mandibular second molar (P2MI).

**Figure 7 ijerph-19-09279-f007:**
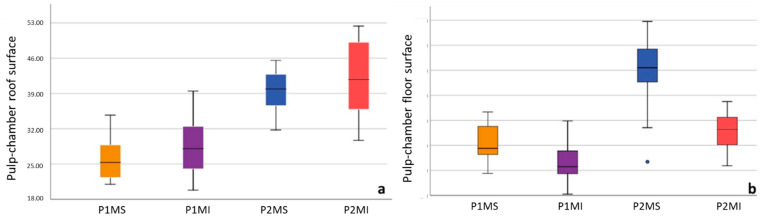
Box plot of pulp chamber surfaces (mm^2^): pulp chamber roof surface (**a**); pulp chamber floor surface (**b**). Outlier data in P2MS box of pulp chamber floor surface (•).

**Figure 8 ijerph-19-09279-f008:**
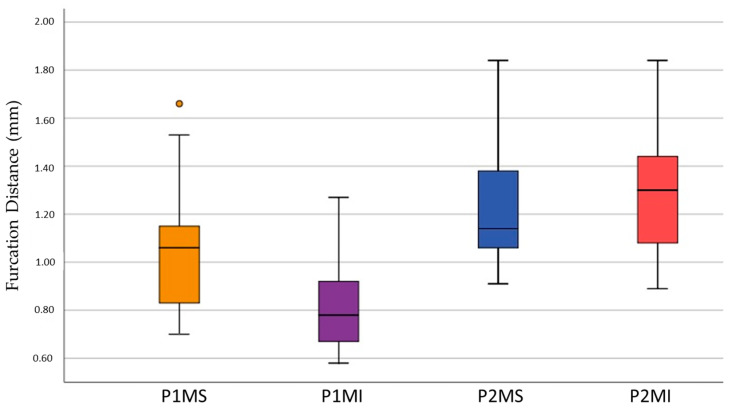
Box plot of distance between pulp chamber floor and furcation (FD). Outlier data in the 1MS box of FD (•).

**Figure 9 ijerph-19-09279-f009:**
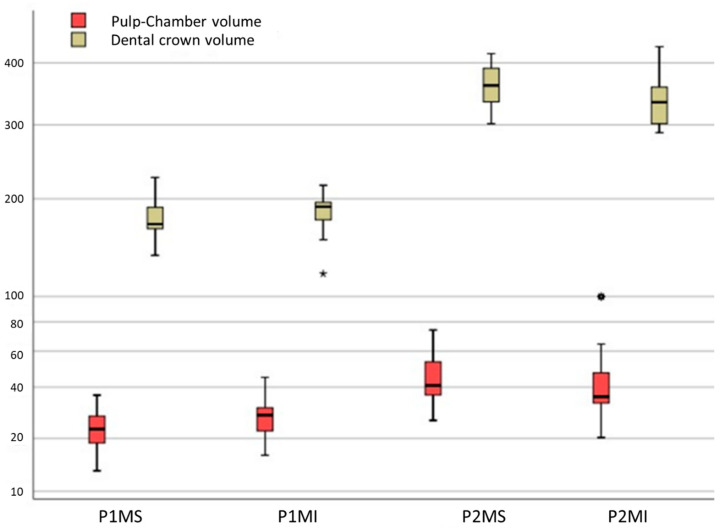
Box plot of pulp chamber (PV) and dental crown volume (CV). Outlier data in 1MI box of CV (*); outlier data in P2MI box of PV (✶).

**Figure 10 ijerph-19-09279-f010:**
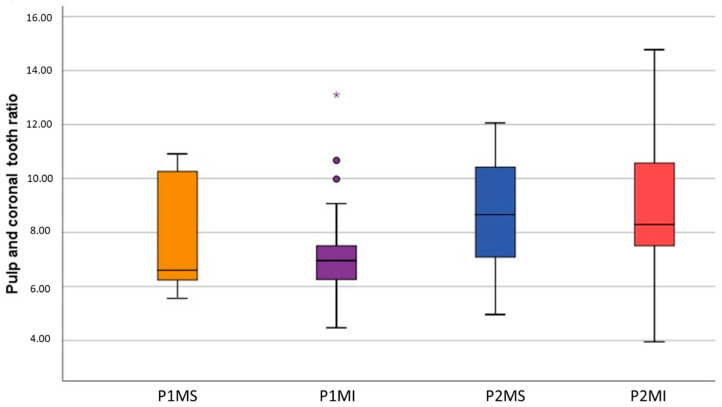
Box plot of tooth and pulp and coronal ratio (VR). Outlier data in P1MI box of VR (∗ • •).

**Figure 11 ijerph-19-09279-f011:**
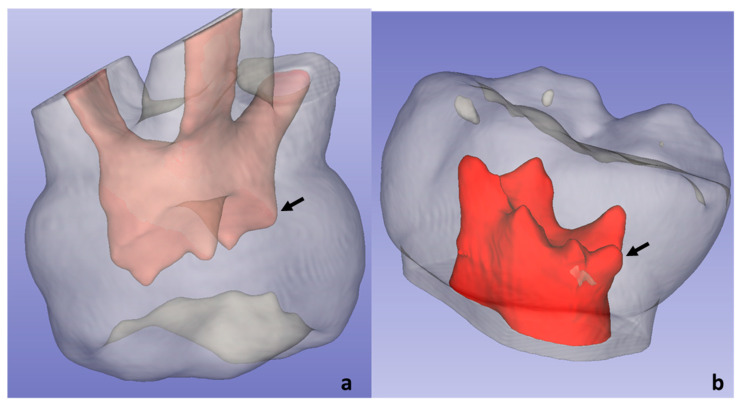
The arrows indicate the accessory pulp horns in P1MS (**a**) and P2MI (**b**).

**Figure 12 ijerph-19-09279-f012:**
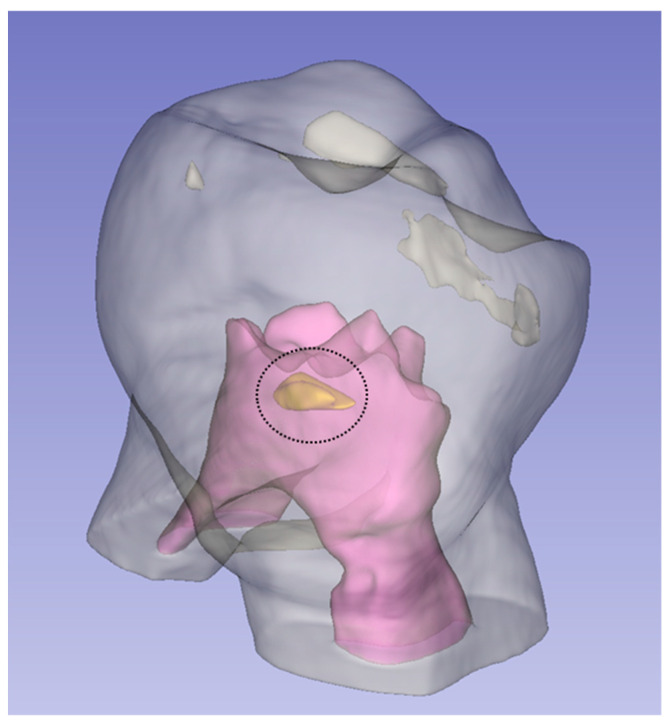
The circumference encloses the three-dimensional hyperdense image reconstruction in 1MI (pulp stone in yellow).

**Table 1 ijerph-19-09279-t001:** Measurements of the pulp chamber, furcation, and coronal portion of the teeth.

Tooth	PCR (mm^2^)		PCF (mm^2^)		FD (mm)		PV (mm^3^)		CV (mm^3^)		VR (%)	
	Mean	SD	Mean	SD	Mean	SD	Mean	SD	Mean	SD	Mean	SD
P1MS	26.07	4.12	10.49	1.61	1.02	0.26	23.22	6.23	173.68	21.77	7.93	1.97
P1MI	28.46	5.25	8.48	1.99	0.81	0.17	26.95	6.89	183.76	20.79	7.18	1.78
P2MS	39.31	4.06	17.34	3.61	1.22	0.27	44.43	12.64	359.90	36.59	8.68	2.25
P2MI	41.88	6.70	11.27	3.93	1.26	0.26	42.19	16.42	338.37	37.15	8.77	2.32
*p*-value *	0.000		0.000		0.000		0.000		0.000		0.014	

Pulp chamber roof (PCR); pulp chamber floor (PCF); distance between pulp chamber floor and furcation (FD); pulp chamber volume (PV); dental crown volume (CV); volume ratio (VR); primary maxillary first molar (P1MS); primary mandibular first molar (P1MI); primary maxillary second molar (P2MS); primary mandibular second molar (P2MI); SD: standard deviation. * Kruskal–Wallis test, statistical significance at *p* ≤ 0.05.

## Data Availability

The datasets generated and analysed during the current study are not publicly available for privacy reasons, but they are available from the corresponding author on reasonable request.

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
