# Peer review of "Three-Dimensional Analysis of the Pulp Chamber and Coronal Tooth of Primary Molars: An In Vitro Study"

_ijerph, 2022, doi:10.3390/ijerph19159279_

Round 1
Reviewer 1 Report
This is an article about the measurement and observation of primary molar using CBCT. Methodological classification is in vitro study.
Please reconsider the following points.
Materials and methods
There is description “The different tissues of tooth were coloured” in line 91. There is no explain about the criteria to classify the different tissues of the tooth objectively.
Chapter 2.4 is missing.
There is no description of the measuring methods of the indexes; CV, PV, PC/CP, PCR, PCF and furcation length. Moreover, it can’t understand the abbreviation, PC/CP.
The abbreviation of the measured items should be mentioned in materials and methods.
At least, the figure legends in Figure 5 is necessary. The meaning of the arrows can not understand.
Results
It is unclear the meaning of DS in table 1.
The meaning of the star and asterisk in figure 9 and 10 is uncertain. There should be a figure legends and explain the meanings.
It is not clear the condition of the “hypodense images” in line 195. The criteria of the accessory canals at furcation should be described in detail.
Discussion
Third paragraph, reference No.1 and 5 were classified as in vivo methodology. The reference No.1 and 5 might be “in vitro” studies using micro-CT.
There is a description of the mixture of the primary molar and molar in line 235.
The meaning of the F/F-D can’t understand without the explanation in line248.
The description “boys and girls” and “male and female” were mixed in line 260 and so on and forth.
Author Response
Dear Reviewer:
Both authors appreciate your comments and suggestions, these have helped improve the way we present our research. Below we will detail and resolve each of your comments.

Reviewer 2 Report
Congratulations on a good performance of your research teem. Introduction and methodology are well described although certain minor spelling or punctuation mistakes were observed. (see lines 30, 118, 122 for example).
In the discussion section though I was quite lost. It is too long and some information reported, should be excluded. (also you don't need to repeat your results (they are mentioned in the
results" section). A more robust discussion section will be appreciated for optimal approval.
Conclusions should also be rewritten in a more specific way (avoid putting numbers again). Their present report diminishes the value of your research.
Thank you and well done!
Author Response

(The authors gave the same response as above.)

Reviewer 3 Report
This article entitled "Three-dimensional analysis of the pulp chamber and coronal tooth of decious molars: an in vitro study" intends to analyze the primary molars at the pulpal level on a three-dimensional basis, carrying out different measurements on them in order to have a broader view of the anatomical differences that can be found.
Introduction
The authors are suggested to improve this introductory part; for example giving an approach to the current situation in this research topic.
Materials and Methods
Within the incussion and exclusion criteria, the authors comment that the degree of physiological wear has been taken into account as a discard factor. How has that degree of wear been calculated?
Have differences in the participants regarding sex, age or dental age been taken into account? Could these items have influenced the results?.
Results
It would be advisable to review tables y figures and specify in it the different initials / acronyms at the bottom of the table.
Discusion
According to the authors, differences have been found between their results and other studies. Which may be due?
Conclusions
In what subjects or areas do the authors consider that their results can be applied? They comment that it could be interesting when it comes to pulp treatments or making preformed crowns. How could it be applied in these areas?
Author Response

(The authors gave the same response as above.)

Round 2
Reviewer 1 Report
The possessive case of inorganic matter should be listed in ”of" in L30, 149, 213 and 336.
It is necessary the mention of the "primary" or " deciduous" in the explanation of 1MS, 2MS, 1MI and 1MI.
Author Response
Dear Reviewer:
Both authors appreciate your comments and suggestions, these have helped improve the way we present our research. Below we will detail and resolve each of your comments.
Response to Reviewer 1 Comments
Point 1: The possessive case of inorganic matter should be listed in ”of" in L30, 149, 213 and 336.
Response 1: The correction is made on the lines indicated.
Previous: “dental crown’s hard tissues [1–3], This fact becomes more important in a primary tooth,”
Modify: hard tissues of dental crown [1–3], This fact becomes more important in a primary tooth
Previous: CBCT as a line or point with less density than hard tissue structure. Their presence was
Modify: CBCT as a line or point with less density than the structure of the hard tissue. Their
Previous: The limits obtained at the 95% confidence interval regarding the pulp volume were
Modify: The limits obtained at the 95% confidence interval regarding the volume of the pulp
Previous: pulp stone was the middle coronal third, a fact to be considered when planning
Modify: pulp stone was the middle of the coronal third, a fact to be considered when planning
Point 2: It is necessary the mention of the "primary" or " deciduous" in the explanation of 1MS, 2MS, 1MI and 1MI.
Response 2: Primary dentition or decidua is added in acronyms (text, tables, headings and figures).